# Excessive Nitrogen Application Leads to Lower Rice Yield and Grain Quality by Inhibiting the Grain Filling of Inferior Grains

Can Zhao, Guangming Liu, Yue Chen, Yan Jiang [ID], Yi Shi, Lingtian Zhao, Pingqiang Liao, Weiling Wang, Ke Xu, Qigen Dai and Zhongyang Huo *

Jiangsu Key Laboratory of Crop Genetics and Physiology, Jiangsu Key Laboratory of Crop Cultivation and Physiology, Jiangsu Co-Innovation Center for Modern Production Technology of Grain Crops, Agricultural College, Yangzhou University, 88 Daxue South Road, Yangzhou 225009, China; canzhao@yzu.edu.cn (C.Z.); 007311@yzu.edu.cn (G.L.); 006900@yzu.edu.cn (Y.C.); mx120200671@yzu.edu.cn (Y.J.); 006809@yzu.edu.cn (Y.S.); mz120190919@yzu.edu.cn (L.Z.); mz120201238@yzu.edu.cn (P.L.); 007465@yzu.edu.cn (W.W.); xuke@yzu.edu.cn (K.X.); qgdai@yzu.edu.cn (Q.D.)
* Correspondence: 003343@yzu.edu.cn

**Abstract:** Nitrogen fertilizer is an important agronomic measure to regulate rice yield and grain quality. Grain filling is crucial for the formation of rice yield and grain quality. However, there are few studies on the effects of excessive nitrogen application (ENA) on grain filling rate and grain quality. A two-year field experiment was conducted to reveal the difference in grain filling characteristics and grain quality of superior grains (SG) and inferior grains (IG), as well as their responses to nitrogen fertilizer. We determined the grain appearance, the rice yield, the grain filling characteristics of SG and IG, and grain quality. We found that with the increasing nitrogen application level, grain yield of both varieties first increased and then decreased. The average yield of excessive nitrogen application (345 kg N ha$^{-1}$) was 2.68–6.31% lower than that of appropriate nitrogen application (270 kg N ha$^{-1}$). ENA reduced the grain filling rate by 12.7–25.8%, and the grain filling rate of SG was higher than that of IG. Increasing nitrogen application increased the processing quality and appearance quality of rice grain, but ENA deteriorated the appearance quality, eating quality and nutritional quality. The amylose content and taste value of SS were 3.1–9.7% and 7.1–20.2% higher than those of IS, respectively. The protein components of SG were lower than those of IG. Taken together, our results revealed that ENA leads to the lowering of rice grain yield and grain quality by suppressed grain filling of inferior grains.

**Keywords:** excessive nitrogen application; grain yield; rice grain quality; superior grains and inferior grains; grain filling

## 1. Introduction

Rice (*Oryza sativa* L.) is a staple food for more than half of the global population, and more than 60% of the total population in China consumes rice as a staple food [1]. It is estimated that by 2030, according to the current rice consumption level, China will need to produce 22% more rice to meet the domestic consumption demand. As one of the main nutrient elements of rice, nitrogen is widely used to improve yield. The production and consumption of nitrogen fertilizer in China ranks first in the world, and the amount of nitrogen fertilizer consumed in China accounts for 30% of the total amount of nitrogen fertilizer applied worldwide [2]. In the main *japonica* rice producing areas of China, the recommend rate of nitrogen application is 240–300 ka N ha$^{-1}$ to obtain a rice yield above 7.5 t ha$^{-1}$. In order to maximize rice yield, farmers in many parts of the world tend to overuse nitrogen fertilizer in the field. Excessive use of nitrogen fertilizer will not only increase production costs and reduce economic benefits, but also cause serious environmental pollution, including acid rain, soil acidification, and water eutrophication [3,4].

The grain-filling stage is crucial for rice yield and quality formation. Grain filling is a stage at which photosynthates are transported from sources to grains in the form of sugars, especially sucrose, forming starch through a series of enzymatic reactions and storing this starch in the grains [5]. The eating quality of grains with different spikelet positions is quite different [6,7]. In general, spikelets that heading early on the primary branch at the top of the panicle are called superior spikelets, while spikelets that heading late on the secondary branch at the base are called inferior spikelets [8]. Compared with superior grains, inferior grains showed lower amylose content and poorer eating quality, but higher total starch content [9,10]. The grain filling characteristics of rice are not only affected by genetic factors, but also different in nitrogen fertilizer, varieties and grain types. Although extensive studies have been performed to reveal the effects of nitrogen application on grain filling, the response of SS and IS to excessive nitrogen application were still poorly understood.

Nitrogen is the important element for rice growth and yield formation. Appropriate nitrogen application could balance rice yield and grain quality, while excessive nitrogen application would deteriorate taste quality [11,12]. Nitrogen fertilizer application can promote the starch biosynthesis and carbohydrate consumption, and has great influence on the grain filling of superior grains and inferior grains. Increasing nitrogen fertilizer can improve the metabolic ability of rice and promote the synthesis of various substances at grain filling stage, thereby affecting the grain formation process [13]. Appropriate nitrogen application can significantly improve the average grain filling rate, maximum grain filling rate and grain weight at the active grain filling stage of inferior grains, thereby increasing the final rice yield [14]. The grain filling rate of low nitrogen treatment was higher than that of high nitrogen treatment [15]. Appropriate nitrogen fertilizer levels can improve the grain filling rate of superior and inferior grains, but there are certain density and variety differences between them. It was reported that the maximum grain filling rate of superior grains increased first and then decreased with the increase in nitrogen fertilizer level, while that of inferior grains showed an increasing trend [16]. Appropriate nitrogen application can not only improve rice yield and grain filling, but also improve grain quality [17]. Grain quality mainly includes the processing quality, appearance quality, nutritional quality and cooking and eating quality [18,19]. Increasing the level of nitrogen fertilizer not only improves the processing and appearance quality, but also improves the protein content and reduces the cooking and eating quality [20]. Grain quality varies greatly among different grain positions. It is reported that increasing panicle fertilizer reduces the processing quality and appearance quality, and mainly affects the grain quality of inferior grains. The appearance quality and eating quality of superior grains are better than those of inferior grains [21,22]. A higher N rate increased the percentages of brown rice and head rice, chalky-kernel rate, but reduced the length/width ratio, chalkiness, apparent amylose content, and gel consistency [11].

Compared with appropriate nitrogen application, excessive nitrogen application reduced rice yield and grain filling rate [23,24], which may be caused by poor grain filling of IS under high nitrogen conditions [7]. There have been many reports about the effect of the nitrogen application rate on the yield and nitrogen use efficiency of rice, but few studies on the differences of grain filling characteristics and grain quality between superior and inferior grains. A two-year paddy field experiment was conducted to reveal the effects of excessive nitrogen application on the grain filling characteristics and grain quality of superior and inferior grains. Our objectives were to display the difference characteristics of grain quality between superior and inferior grains and their response to nitrogen fertilizer level, and to clarify the relationship between grain quality and grain filling. The outcomes of this study provide useful information for rice production to achieve higher grain yield and higher grain quality, and provided a theoretical basis for the rational application of nitrogen fertilizer to improve the grain quality of single season late *japonica* rice in the Jiangsu province.

## 2. Materials and Methods

### 2.1. Plant Materials and Experimental Design

Field experiments were conducted in a local farmer's paddy field in Shatou town (119°35′ E, 32°26′ N), Guangling District, Yangzhou City, Jiangsu Province in the summer cropping season (between May and November) of 2019 and 2020. The field soil was a sandy loam, which contained 21.01 g kg$^{-1}$ organic matter, 157.23 mg kg$^{-1}$ alkaline hydrolysis nitrogen, 1.42 g kg$^{-1}$ total N, 18.67 mg kg$^{-1}$ Olsen-P, 138.55 mg kg$^{-1}$ available K, and the soil pH was 7.79. The region is located in central Jiangsu Province and has a humid subtropical climate. The experiment was arranged in a split plot design, *Japonica* rice cultivar Nanjing 9108 and Ningjing 7 were used as material. Both NJ9108 and NJ7 are super rice varieties widely planted in the central region of Jiangsu province, both of which are *japonica* rice. They have almost the same heading date and growth period when planted in Yangzhou. The heading dates of NJ9108 are 26 August 2019 and 28 August 2020, and the whole growth period is 151 days. The heading dates of NJ7 are 27 August 2019 and 29 August 2020, and the whole growth period is 153 days. Four fertilization treatments were arranged: 0 (CK), 195 kg·ha$^{-1}$, 270 kg·ha$^{-1}$ and 345 kg·ha$^{-1}$, which were represented by N1 (0N), N2 (low nitrogen), N3 (appropriate nitrogen), and N4 (excessive nitrogen), respectively. Seeds were sown in plastic plates on 25 May in both 2019 and 2020, with a seeding rate of 120 g of dry seeds per plate. Seedlings were manually transplanted in hills on 14 June of each year. The area of each experimental plot was 20 m$^2$, with 50 cm spaces between the adjacent plots, and hill spacing was 12 cm × 30 cm with four seedlings per hill. Each plot was separated by a soil ridge (35 cm wide and 20 cm high) and covered with a plastic film. The individual plants of each accession were planted in separate plots with three replications. Nitrogen fertilizer was distributed at a ratio of 4:2:2:2 (6:4) as basic fertilizer, tillering fertilizer, spikelet-promoting fertilizer and spikelet-protecting fertilizer, in which spikelet-promoting fertilizer and spikelet-protecting fertilizer were applied at the 4th and 2nd leaf-age from top, respectively. Calcium superphosphate (P$_2$O$_5$ content: 12%) and potassium chloride (K$_2$O content: 60%) were applied as basal at the rates of 150 kg P$_2$O$_5$ ha$^{-1}$ and 240 kg K$_2$O ha$^{-1}$, respectively. Insect pests, pathogens and weeds were controlled using common chemical treatments.

### 2.2. Sampling and Determination of Rice Yield and Grain-Filling Rate

According to Zhao's method [4], at maturity stage, rice yield was determined from all plants in a 6.0 m$^2$ area (except for border plants) in each plot and was calculated based on a standardized moisture content of 14%.

A total of 320–340 panicles that headed on the same day were chosen and tagged for each plot. The flowering date of each upper spikelet on the tagged panicles was recorded. At every 5th day from flowering to harvest, 15–20 marked panicles of each cultivar in each treatment were sampled by picking superior grains and inferior grains, which were then placed in an oven at 105 °C for 30 min, and dry it at 80 °C to constant weight. According to the method of Ishimaru et al. [8], superior and inferior spikelets were separated, then removed glumes and weighed. The processes of grain filling were fitted with Richards' growth equation [25] as described by Zhu et al. [26]:

$$W = A/(1 + Be^{-kt})^{1/N} \tag{1}$$

$$G = (KW/N)[1 - (W/A)^N] \tag{2}$$

$$T_{G.max} = (\ln B - \ln N)/K \tag{3}$$

$$G = AK/[2(N + 2)] \tag{4}$$

$$D = 2(N + 2)/K \tag{5}$$

Put the $T_{G.max}$ into Equations (1) and (2), and we can find the $G_{max}$, where $W$ is the grain weight (mg), $A$ is the final grain weight, $t$ is the time after anthesis, and B, k, K, and

N are coefficients determined by regression, $G$ is the grain-filling rate, $T_{G.max}$ is time of reaching the maximum grain-filling rate, $G_{max}$ is the maximum grain-filling rate, $G$ is the mean grain-filling rate, and $D$ is the active grain-filling period.

### 2.3. Determination of Milling Quality and Appearance Quality of Rice Grain

The methods for determination of milling quality and appearance quality were modified from those described by Huang et al. [27]. The chalky grain rate and chalkiness degree were calculated as follows:

$$\text{Chalky grain rate (\%)} = \text{Number of chalky grains}/\text{Number of observed grains} \times 100 \quad (6)$$

$$\text{Chalkiness degree (\%)} = \text{Chalky area}/\text{Total area of observed grains} \times 100 \quad (7)$$

The relevant indicators of milling quality are calculated as follows:

$$\text{Brown rice rate (\%)} = \text{Weight of brown rice} \times 100/\text{Weight of rice husk and grain} \quad (8)$$

$$\text{Milled rice ratio (\%)} = \text{Weight of milled rice} \times 100/\text{Weight of rice husk and grain} \quad (9)$$

$$\text{Head rice yield (\%)} = \text{Weight of head rice} \times 100/\text{Weight of rice husk and grain} \quad (10)$$

### 2.4. Determination of Eating Quality

Eating-quality score was measured in a Cooked Rice Taste Analyzer STA1A (Satake Co., Ltd., Hiroshima, Japan) according to the Mikami' methods [28]. The gel consistency of the rice grains was determined according to the method illustrated by Rayee et al. [29] with fewer modifications. Briefly, in clean and dried test tubes, 100 mg of fine rice powder in triplicate were taken followed by adding of 0.2 mL of ethanol (95%) containing thymol blue to prevent the rice powder from clumping. The mixture was vortexed slowly, and 2 mL of potassium hydroxide (KOH) (0.2N) was added and mixed the contents properly. All the samples were kept in a boiling water bath for 8 min then cooled for 5 min. All the samples were vortexed again and kept in an ice water bath for 20 min. Later on, all the tubes were taken out and laid horizontally on laminated graph paper for one hour to take the measurement. The length of gel was determined from the lowest part of the tube to the end of the gel, expressed in mm.

### 2.5. Determination of Amylose Content and Protein Composition

The amylose content was determined as described by Wang et al. [30]. Starch was defatted in methanol/water (85:15, *v/v*) and then dissolved in dimethyl sulphoxide containing a urea (UDMSO) solution. The starch-UDMSO solution was treated with an $I_2$-KI solution. The iodine absorption spectrum was scanned from 400 to 900 nm using an Ultrospec 6300 Pro spectrophotometer (Amersham Biosciences, Amersham, UK). Iodine blue was measured at 680 nm and the amylose content was calculated from the absorbance at 620 nm by referencing a standard curve prepared with amylopectin from corn and amylose from potato. The protein content was measured from the nitrogen content using the Kjeldahl method with a conversion coefficient of 5.95 [31].

### 2.6. Data Analysis

Multivariate analyses of variance (MANOVA) were conducted to determine the effects of year and variety and treatment as well as their interaction effects on the yield and yield's components at a significance level of 5%. Data were tested for normality (Shapiro-Wilk test, $p > 0.05$) and homogeneity of variance (Levene's test, $p > 0.05$) before MANOVAs. When comparing the four nitrogen application rates, Duncan's multiple range test ($p < 0.05$) was used. For superior and inferior grains, the means were compared using the independent-sample $t$-test ($p < 0.05$). All statistical analyses were conducted using the SPSS software package (18.0; SPSS Inc., Chicago, IL, USA). The values in the figures and tables are presented as mean $\pm$ standard error (SE).

## 3. Results

### 3.1. Rice Yield and Its Components

The analysis of variance showed that there were extremely significant differences in 1000-grain weight, seed setting rate and rice yield between 2019 and 2020. Year (Y) and treatment (T) significantly affected grain yield without interaction. The panicle number, grains number per panicle and seed setting rate had extremely significant interaction effects between years and varieties. The interactions of Y × T and V × T were highly significant in rice yield components. In both 2019 and 2020, the rice yield increased and then decreased with the increasing nitrogen fertilizer application. The yield of NJ9108 under low nitrogen condition (N1, N2) is lower than that of NJ7, and the yield of NJ9108 under high nitrogen condition (N3, N4) is similar to that of NJ7. The yield of N3 was the highest, and the yield of excessive nitrogen application (N4) decreased. Compared with N1, the yield of N3 increased from 67.6 to 85.1%, across 2019 and 2020. Excessive nitrogen application did not strongly reduced grain yield, but it tended to be lower yield when compared with N3. The mean yield of two varieties under excessive nitrogen application (N4) was 2.7–6.3% lower than that under moderate nitrogen application (N3) (Table 1). Increasing nitrogen fertilizer had significant effect on yield components. The panicle number and grain number per panicle of N4 were 45.2–78.6% and 7.2–31.0% higher than those of N1, respectively. In 2019 and 2020, the 1000-grain weight of the two varieties decreased with the increase in nitrogen application rate, and N4 decreased by 7.1–11.5% compared with N1. Grains per panicle in NJ9108 was lower than that in NJ7 and seed setting rate in NJ9108 was larger than that in NJ7 across 2019 and 2020. There was significant difference in seed setting rate among different treatments, which decreased with the increase in nitrogen application rate.

**Table 1.** Effect of nitrogen fertilizer level on grain yield and its components.

| Year | Variety | Treatment | Grain Yield (t ha$^{-1}$) | Panicle Number (×10$^4$ ha$^{-1}$) | Grains per Panicle | 1000-Grain Weight (g) | Seed Setting Rate (%) |
|---|---|---|---|---|---|---|---|
| 2019 | NJ9108 | N1 | 6.1 ± 0.18 c | 188 ± 4.06 c | 131 ± 3.42 c | 28.2 ± 0.03 a | 98.2 ± 0.24 a |
| | | N2 | 9.8 ± 0.13 b | 286 ± 1.45 b | 132 ± 0.64 bc | 27.4 ± 0.15 b | 95.7 ± 0.39 b |
| | | N3 | 11.4 ± 0.10 a | 330 ± 5.37 a | 137 ± 1.26 ab | 27.1 ± 0.10 b | 95.3 ± 0.45 b |
| | | N4 | 10.9 ± 0.08 a | 335 ± 1.00 a | 140 ± 1.01 a | 26.6 ± 0.24 c | 89.9 ± 0.44 c |
| | | Mean | 9.6 | 285.00 | 135.0 | 27.3 | 94.8 |
| | | CV (%) | 22.6 | 21.8 | 3.6 | 2.3 | 3.4 |
| | NJ7 | N1 | 6.8 ± 0.07 c | 208 ± 6.59 d | 157 ± 1.52 c | 27.4 ± 0.21 a | 94.8 ± 0.15 a |
| | | N2 | 10. ± 0.08 b | 295 ± 2.05 c | 161 ± 2.67 b | 26.8 ± 0.08 b | 90.9 ± 0.39 b |
| | | N3 | 11.4 ± 0.07 a | 326 ± 0.30 b | 169 ± 0.66 a | 26.0 ± 0.10 c | 83.7 ± 0.33 c |
| | | N4 | 10.8 ± 0.12 ab | 339 ± 4.60 a | 172 ± 0.63 a | 25.1 ± 0.04 d | 81.4 ± 0.26 d |
| | | Mean | 9.8 | 292.00 | 165.00 | 26.4 | 87.7 |
| | | CV (%) | 19.2 | 18.4 | 4.3 | 3.4 | 6.4 |
| 2020 | NJ9108 | N1 | 6.0 ± 0.08 c | 216 ± 0.81 c | 126 ± 2.02 c | 27.6 ± 0.02 a | 96.2 ± 0.49 a |
| | | N2 | 9.5 ± 0.09 b | 284 ± 0.20 b | 133 ± 1.26 b | 26.7 ± 0.09 b | 95.1 ± 0.22 b |
| | | N3 | 11.0 ± 0.03 a | 324 ± 0.20 a | 144 ± 0.65 a | 26.1 ± 0.10 c | 94.7 ± 0.27 b |
| | | N4 | 10.9 ± 0.01 a | 331 ± 0.24 a | 145 ± 0.89 a | 25.8 ± 0.05 d | 90.7 ± 0.50 c |
| | | Mean | 9.4 | 289.00 | 137.00 | 26.6 | 94.2 |
| | | CV (%) | 22.4 | 16.4 | 6.3 | 2.8 | 2.4 |
| | NJ7 | N1 | 6.4 ± 0.65 c | 220.0 ± 4.18 c | 141 ± 1.25 c | 27.2 ± 0.05 a | 95.4 ± 0.26 a |
| | | N2 | 9.7 ± 0.10 b | 285 ± 1.47 b | 165 ± 0.43 b | 25.5 ± 0.07 b | 92.6 ± 0.23 b |
| | | N3 | 10.8 ± 0.57 a | 310 ± 5.00 a | 168 ± 0.92 ab | 24.8 ± 0.04 c | 89.7 ± 0.34 c |
| | | N4 | 10.0 ± 0.57 ab | 320 ± 5.00 a | 172 ± 0.24 a | 24.4 ± 0.04 d | 84.6 ± 0.34 d |
| | | Mean | 9.3 | 284.00 | 161.5 | 25.5 | 90.6 |
| | | CV (%) | 21.0 | 14.5 | 7.9 | 4.4 | 4.6 |

**Table 1.** *Cont.*

| Year | Variety | Treatment | Grain Yield (t ha$^{-1}$) | Panicle Number (×10$^4$ ha$^{-1}$) | Grains per Panicle | 1000-Grain Weight (g) | Seed Setting Rate (%) |
|---|---|---|---|---|---|---|---|
| | | Year | ** | ns | ns | ** | ** |
| | | Variety | ns | ns | ** | ** | ** |
| | Analysis of variance | Treatment | ** | ** | ** | ** | ** |
| | | Y × V | ns | ** | ** | ns | ** |
| | | Y × T | ns | ** | ** | ** | ** |
| | | V × T | ns | ** | ** | ** | ** |
| | | Y × V × T | ns | ns | * | * | ** |

Different letters indicate statistical significance at *p* = 0.05 within the same column, year and variety. "CV" means the coefficient of variation. N1, 0 kg N·ha$^{-1}$; N2, 195 kg N·ha$^{-1}$; N3, 270 kg N·ha$^{-1}$; N4, 345 kg N·ha$^{-1}$. * and ** mean significant difference at the 0.05 and 0.01 probability levels, respectively; ns, not significant (*p* > 0.05).

### 3.2. Grain Processing Quality

Increasing nitrogen fertilizer can improve grain processing quality. In 2019 and 2020, with the increase in nitrogen application rate, the brown rice rate, milled rice rate and head milled rice rate gradually increased (Table 2). The processing quality of SG was better than that of IG. The brown rice rate, milled rice rate and head rice rate of SG were 1.12–4.37%, 0.52–6.39% and 3.70–8.10% higher than those of IG, respectively, across 2019 and 2020. The coefficient of variation (CV) of the processing quality of the IG was higher than that of SG (Table 2).

**Table 2.** Effects of nitrogen fertilizer levels on processing quality of superior and inferior grains in 2019 and 2020.

| Year | Variety | Category | Treatment | Brown Rice Rate | Milled Rice Rate | Head Rice Rate |
|---|---|---|---|---|---|---|
| 2019 | NJ9108 | SG | N1 | 82.6 ± 0.30 Ac | 71.0 ± 0.27 Ac | 64.4 ± 1.11 Ac |
| | | | N2 | 84.0 ± 0.07 Ab | 73.5 ± 0.23 Ab | 67.8 ± 0.49 Ab |
| | | | N3 | 85.2 ± 0.08 Aa | 74.4 ± 0.18 Ab | 69.9 ± 0.10 Aab |
| | | | N4 | 85.9 ± 0.07 Aa | 75.8 ± 0.27 Aa | 71.5 ± 0.10 Aa |
| | | | CV (%) | 1.58 | 2.57 | 4.24 |
| | | IG | N1 | 80.2 ± 0.29 Bc | 70.2 ± 0.06 Bd | 60.9 ± 0.69 Bc |
| | | | N2 | 81.4 ± 0.33 Bb | 71.6 ± 0.29 Bc | 62.5 ± 0.64 Bbc |
| | | | N3 | 82.7 ± 0.06 Ba | 73.5 ± 0.06 Bb | 63.6 ± 0.59 Bb |
| | | | N4 | 82.4 ± 0.27 Bab | 75 ± 0.28 Ba | 66.7 ± 0.38 Ba |
| | | | CV (%) | 1.34 | 2.66 | 3.76 |
| | NJ7 | SG | N1 | 82.7 ± 0.30 Ab | 70.0 ± 0.06 Ac | 65.0 ± 0.26 Ac |
| | | | N2 | 84.2 ± 0.06 Aa | 71.7 ± 0.10 Ab | 67.4 ± 0.21 Ab |
| | | | N3 | 84.6 ± 0.09 Aa | 72.5 ± 0.06 Ab | 68.0 ± 0.20 Ab |
| | | | N4 | 85.2 ± 0.09 Aa | 74.0 ± 0.30 Aa | 70.6 ± 0.18 Aa |
| | | | CV (%) | 1.2 | 2.1 | 3.11 |
| | | IG | N1 | 78.2 ± 0.07 Bd | 70.3 ± 0.31 Ac | 60.4 ± 0.14 Bc |
| | | | N2 | 79.7 ± 0.27 Bc | 71.7 ± 0.30 Ab | 61.9 ± 0.11 Bbc |
| | | | N3 | 81.5 ± 0.28 Bb | 71.5 ± 0.29 Bb | 63.8 ± 0.23 Bab |
| | | | N4 | 83.5 ± 0.27 Ba | 73.5 ± 0.06 Aa | 65.3 ± 0.40 Ba |
| | | | CV (%) | 2.62 | 1.75 | 3.14 |
| 2020 | NJ9108 | SG | N1 | 82.0 ± 0.16 Ad | 72.9 ± 0.06 Ac | 69.4 ± 0.15 Ad |
| | | | N2 | 83.4 ± 0.05 Ac | 74.5 ± 0.20 Ab | 71.8 ± 0.24 Ac |
| | | | N3 | 84.5 ± 0.35 Ab | 75.8 ± 0.08 Aa | 72.7 ± 0.17 Ab |
| | | | N4 | 85.2 ± 0.04 Aa | 76.8 ± 0.19 Aa | 73.9 ± 0.30 Aa |
| | | | CV (%) | 1.51 | 2.06 | 2.4 |
| | | IG | N1 | 81.7 ± 0.06 Ab | 70.7 ± 0.08 Bc | 67.3 ± 0.18 B |
| | | | N2 | 82.4 ± 0.13 Bb | 71.8 ± 0.25 Bc | 68.0 ± 0.33 B |
| | | | N3 | 83.3 ± 0.13 Ba | 73.2 ± 0.29 Bb | 69.4 ± 0.25 B |
| | | | N4 | 84.0 ± 0.24 Ba | 76.0 ± 0.17 Ba | 72.9 ± 0.20 B |
| | | | CV (%) | 1.12 | 2.84 | 3.31 |

**Table 2.** *Cont.*

| Year | Variety | Category | Treatment | Brown Rice Rate | Milled Rice Rate | Head Rice Rate |
|---|---|---|---|---|---|---|
| | NJ7 | SG | N1 | 82.9 ± 0.10 Ac | 74.8 ± 0.08 Ad | 71.4 ± 0.07 Ad |
| | | | N2 | 83.7 ± 0.16 Ab | 75.3 ± 0.06 Ac | 72.3 ± 0.24 Ac |
| | | | N3 | 84.6 ± 0.07 Aa | 75.8 ± 0.09 Ab | 73.3 ± 0.07 Ab |
| | | | N4 | 84.5 ± 0.03 Aa | 76.6 ± 0.06 Aa | 74.5 ± 0.04 Aa |
| | | | CV (%) | 0.89 | 0.94 | 1.64 |
| | | IG | N1 | 80.1 ± 0.35 Bc | 68.6 ± 0.33 Bb | 64.4 ± 0.31 B |
| | | | N2 | 81.9 ± 0.11 Bb | 70.0 ± 0.26 Bab | 65.9 ± 0.53 B |
| | | | N3 | 83.2 ± 0.08 Bab | 72.4 ± 0.72 Bab | 68.7 ± 0.19 B |
| | | | N4 | 83.9 ± 0.11 Ba | 73.4 ± 0.51 Ba | 70.7 ± 0.21 B |
| | | | CV (%) | 1.83 | 2.98 | 3.84 |

Different lowercase letters indicate statistically significant differences among nitrogen application rates according to a Duncan's multiple range test ($p < 0.05$). Different capital letters indicate statistical significance for the comparison between superior and inferior grains under the same nitrogen application rate (independent-sample $t$-test, $p < 0.05$). N1, 0 kg N·ha$^{-1}$; N2, 195 kg N·ha$^{-1}$; N3, 270 kg N·ha$^{-1}$; N4, 345 kg N·ha$^{-1}$. CV, the coefficient of variation. SG, the superior grain; IG, the inferior grain.

### 3.3. Rice Grain Morphology and Appearance Quality

With the increase in nitrogen application rate, the appearance of the two varieties became worse. Under 0N (N1), SG and IG were dark yellow, with a smooth surface, and full grains, which had little difference. Under excessive nitrogen application (N4), SG and IG were significantly different, especially the surface of IG was wrinkled and shriveled, the grains were small and slender, and showed green (Figure 1A). An appropriate amount of (N2–N3) nitrogen fertilizer could improve the appearance of the head milled rice of SG and IG, as well as reducing the chalkiness making grains more transparent. Compared with SG, the morphology of IG under excessive nitrogen application (N4) became worse, with slender and chalkier grains (Figure 1B).

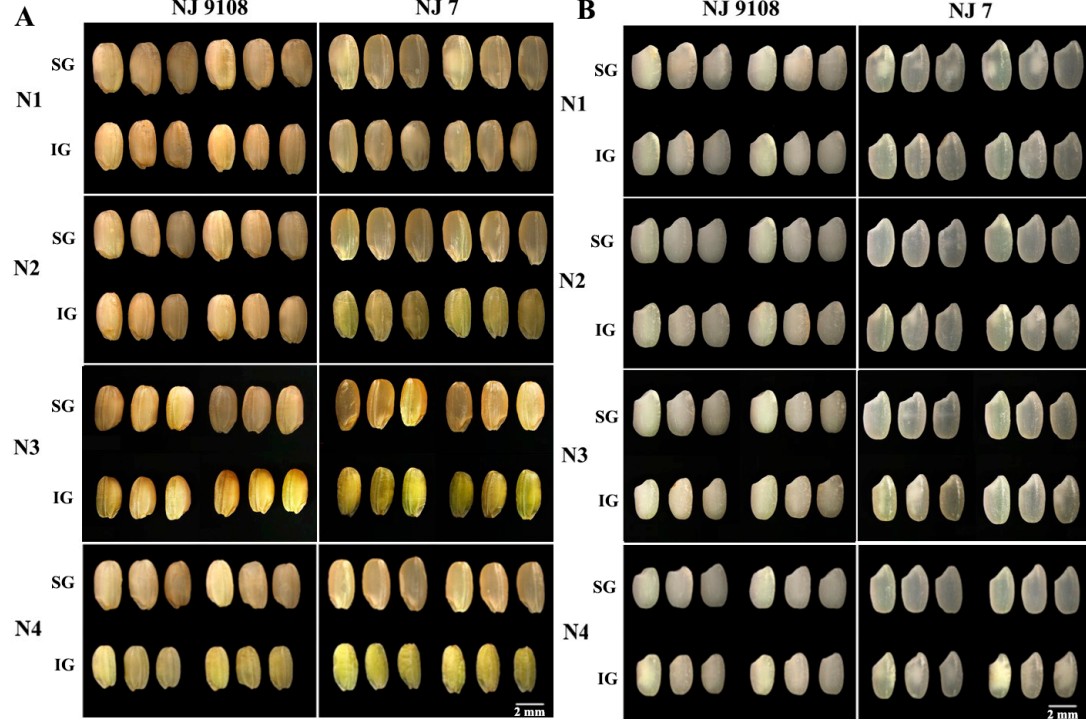

**Figure 1.** Morphological changes of superior and inferior grains under different nitrogen fertilizer levels. (**A**), brown rice; (**B**), milled rice. SG, the superior grain; IG, the inferior grain. N1: 0 kg N·hm$^{-2}$, N2: 195 kg N·hm$^{-2}$, N3: 270 kg N·hm$^{-2}$, N4: 345 kg N·hm$^{-2}$.

With the increase in nitrogen fertilizer level, the chalky rate, chalkiness area and chalkiness of SG and IG decreased first and then increased, and the appearance quality was the best under N3. Compared with N1, the chalky rate, chalkiness area and chalkiness of SG and IG under N3 decreased on average by 25.9–28.3%, 33.6–40.0% and 51.7–65.1%, respectively. The appearance quality of the SG of the two varieties was better than that of the IG (Table 3). The chalky rate, chalkiness area and chalkiness of IG were 23.31–39.72%, 19.91–83.15% and 44.46–133.83% higher than those of SG, respectively, across 2019 and 2020 (Table 3). It was worth mentioning that the chalky rate, chalkiness area and chalkiness of excessive nitrogen application (N4) were lower than those of appropriate nitrogen application (N3). Increasing nitrogen application slightly reduced the grain length and width of SG and IG, and increased the grain length-width ratio. The grain length of the N4 was 1.1–2.5% shorter than that of N1 across 2019 and 2020. The grain width of the N4 was 1.8–7.9% narrower than that of N1 across 2019 and 2020. In the two years, the CV of grain length, grain width and the grain length-width ratio of IG of the two varieties were higher than those of SG, indicating that the grain morphology of IG was more easily affected by nitrogen application rate (Table 3).

**Table 3.** Effects of nitrogen fertilizer levels on appearance quality of strong and weak grains in rice in 2019 and 2020.

| Year | Variety | Category | Treatment | Chalky Rate (%) | Chalkiness Area (%) | Chalkiness (%) | Grain Length (mm) | Grain Width (mm) | Length-Width Ratio |
|---|---|---|---|---|---|---|---|---|---|
| 2019 | NJ9108 | SG | N1 | 51.2 ± 1.13 Ba | 19.4 ± 0.50 Ba | 9.97 ± 0.46 Ba | 4.64 ± 0.01 Aa | 2.88 ± 0.01 Aa | 1.60 ± 0.00 Bb |
| | | | N2 | 47.6 ± 5.39 Ba | 17.9 ± 0.72 Bab | 8.54 ± 1.12 Ba | 4.62 ± 0.00 Aa | 2.86 ± 0.00 Ab | 1.61 ± 0.00 Bab |
| | | | N3 | 25.4 ± 2.94 Bb | 13.9 ± 0.50 Bc | 3.53 ± 0.30 Bb | 4.58 ± 0.01 Ab | 2.83 ± 0.00 Ac | 1.61 ± 0.00 Bab |
| | | | N4 | 29.5 ± 0.97 Bb | 15.5 ± 1.05 Bbc | 4.58 ± 0.15 Bb | 4.56 ± 0.01 Ab | 2.81 ± 0.00 Ad | 1.62 ± 0.00 Ba |
| | | | CV (%) | 31.04 | 13.78 | 42.67 | 0.72 | 0.92 | 0.39 |
| | | IG | N1 | 54.0 ± 0.66 Aa | 24.0 ± 5.54 Aa | 13.0 ± 2.86 Aa | 4.62 ± 0.00 Aa | 2.78 ± 0.02 Ba | 1.66 ± 0.01 Ac |
| | | | N2 | 50.3 ± 1.48 Aa | 21.2 ± 5.11 Aab | 10.6 ± 2.27 Ab | 4.60 ± 0.01 Aa | 2.74 ± 0.04 Ba | 1.68 ± 0.02 Ac |
| | | | N3 | 41.4 ± 2.46 Ab | 16.1 ± 1.07 Ac | 6.67 ± 0.40 Ac | 4.56 ± 0.02 Ab | 2.63 ± 0.02 Bb | 1.72 ± 0.00 Ab |
| | | | N4 | 44.1 ± 7.22 Ab | 18.6 ± 0.50 Abc | 8.23 ± 1.37 Ac | 4.52 ± 0.05 Bb | 2.56 ± 0.04 Bc | 1.76 ± 0.01 Aa |
| | | | CV (%) | 12.96 | 22.42 | 31.26 | 1.1 | 3.55 | 2.57 |
| | NJ7 | SG | N1 | 30.6 ± 0.82 Ba | 12.0 ± 1.12 Ba | 3.69 ± 0.43 Ba | 4.95 ± 0.00 Aa | 2.77 ± 0.00 Aa | 1.78 ± 0.00 Bc |
| | | | N2 | 24.6 ± 1.36 Bb | 11.1 ± 0.24 Ba | 2.76 ± 0.12 Bab | 4.93 ± 0.00 Ab | 2.74 ± 0.00 Ab | 1.79 ± 0.00 Bbc |
| | | | N3 | 19.8 ± 2.55 Bb | 8.85 ± 0.45 Ba | 1.75 ± 0.25 Bb | 4.92 ± 0.00 Ac | 2.70 ± 0.00 Ac | 1.81 ± 0.00 Bb |
| | | | N4 | 21.4 ± 2.01 Bb | 10.2 ± 0.88 Ba | 2.18 ± 0.03 Bab | 4.89 ± 0.00 Ad | 2.64 ± 0.02 Ad | 1.84 ± 0.02 Ba |
| | | | CV (%) | 18.9 | 13.22 | 30.35 | 0.5 | 1.81 | 1.4 |
| | | IG | N1 | 36.9 ± 1.73 Aa | 23.3 ± 0.20 Aa | 8.62 ± 0.48 Aa | 4.84 ± 0.01 Ba | 2.62 ± 0.01 Ba | 1.84 ± 0.01 Ac |
| | | | N2 | 30.9 ± 0.48 Ab | 20.4 ± 1.66 Aab | 6.33 ± 0.41 Ab | 4.80 ± 0.01 Bb | 2.58 ± 0.00 Bb | 1.86 ± 0.00 Ac |
| | | | N3 | 26.7 ± 0.59 Ab | 15.8 ± 0.64 Ac | 4.23 ± 0.22 Ac | 4.74 ± 0.01 Bc | 2.49 ± 0.02 Bc | 1.90 ± 0.01 Ab |
| | | | N4 | 28.5 ± 2.82 Ab | 17.8 ± 1.75 Abc | 5.12 ± 0.93 Abc | 4.71 ± 0.00 Bd | 2.42 ± 0.02 Bd | 1.94 ± 0.02 Aa |
| | | | CV (%) | 13.86 | 16.22 | 29.49 | 1.14 | 3.24 | 2.24 |
| 2020 | NJ9108 | SG | N1 | 42.1 ± 2.94 Ba | 22.5 ± 0.63 Ba | 9.50 ± 0.53 Ba | 4.65 ± 0.00 Aa | 2.85 ± 0.00 Aa | 1.62 ± 0.00 Bb |
| | | | N2 | 32.3 ± 2.59 Bb | 20.7 ± 0.49 Bb | 6.72 ± 0.69 Bb | 4.63 ± 0.00 Aab | 2.83 ± 0.00 Aa | 1.63 ± 0.00 Bb |
| | | | N3 | 23.9 ± 0.56 Bc | 14.2 ± 1.03 Bd | 3.40 ± 0.17 Bd | 4.61 ± 0.00 Ab | 2.82 ± 0.00 Aa | 1.63 ± 0.00 Bb |
| | | | N4 | 25.1 ± 2.46 Bc | 16.8 ± 0.91 Bc | 4.24 ± 0.48 Bc | 4.60 ± 0.00 Ab | 2.78 ± 0.00 Ab | 1.65 ± 0.00 Ba |
| | | | CV (%) | 25.34 | 18.66 | 42.26 | 0.43 | 0.98 | 0.68 |
| | | IG | N1 | 50.7 ± 0.69 Aa | 29.2 ± 0.86 Aa | 14.8 ± 0.63 Aa | 4.45 ± 0.03 Ba | 2.62 ± 0.04 Ba | 1.69 ± 0.01 Ab |
| | | | N2 | 46.2 ± 1.65 Ab | 22.0 ± 0.44 Ab | 10.1 ± 0.50 Ab | 4.43 ± 0.01 Bab | 2.59 ± 0.00 Bb | 1.71 ± 0.00 Aa |
| | | | N3 | 36.7 ± 2.54 Ac | 17.7 ± 0.41 Ad | 6.51 ± 0.56 Ad | 4.42 ± 0.01 Bab | 2.58 ± 0.02 Bbc | 1.71 ± 0.01 Aa |
| | | | N4 | 38.9 ± 0.41 Ac | 19.9 ± 0.49 Ac | 7.78 ± 0.16 Ac | 4.40 ± 0.00 Bb | 2.55 ± 0.00 Bc | 1.72 ± 0.00 Aa |
| | | | CV (%) | 13.88 | 20.42 | 34.03 | 0.55 | 1.34 | 0.84 |
| | NJ7 | SG | N1 | 28.4 ± 0.92 Ba | 21.0 ± 0.45 Ba | 5.98 ± 0.07 Ba | 5.03 ± 0.00 Aa | 2.76 ± 0.00 Aa | 1.81 ± 0.00 Ba |
| | | | N2 | 23.5 ± 0.43 Bb | 18.7 ± 0.43 Bb | 4.40 ± 0.14 Bb | 5.01 ± 0.00 Aab | 2.75 ± 0.00 Aab | 1.82 ± 0.00 Ba |
| | | | N3 | 17.0 ± 1.55 Bc | 10.1 ± 0.54 Bd | 1.73 ± 0.10 Bd | 4.98 ± 0.00 Abc | 2.73 ± 0.00 Abc | 1.82 ± 0.00 Ba |
| | | | N4 | 21.1 ± 1.78 Bb | 13.6 ± 0.44 Bc | 2.89 ± 0.34 Bc | 4.96 ± 0.02 Ac | 2.71 ± 0.01 Ac | 1.82 ± 0.01 Ba |
| | | | CV (%) | 19.66 | 28.01 | 44.68 | 0.58 | 0.76 | 0.45 |
| | | IG | N1 | 32.6 ± 3.96 Aa | 27.3 ± 0.95 Aa | 8.91 ± 1.07 Aa | 4.83 ± 0.03 Ba | 2.54 ± 0.00 Ba | 1.89 ± 0.01 Ad |
| | | | N2 | 29.3 ± 0.57 Ab | 23.7 ± 0.22 Ab | 6.98 ± 0.20 Ab | 4.81 ± 0.01 Ba | 2.50 ± 0.01 Bb | 1.92 ± 0.01 Ac |
| | | | N3 | 24.4 ± 0.57 Ac | 16.8 ± 0.46 Ad | 4.12 ± 0.19 Ac | 4.74 ± 0.00 Bb | 2.45 ± 0.00 Bc | 1.93 ± 0.00 Ab |
| | | | N4 | 25.5 ± 1.16 Ac | 18.9 ± 0.70 Ac | 4.85 ± 0.39 Ac | 4.71 ± 0.06 Bb | 2.38 ± 0.04 Bd | 1.97 ± 0.01 Aa |
| | | | CV (%) | 13.71 | 19.78 | 32.57 | 1.28 | 2.78 | 1.66 |

Different lowercase letters indicate statistically significant differences among nitrogen application rates according to a Duncan's multiple range test ($p < 0.05$). Different capital letters indicate statistical significance for the comparison between superior and inferior grains under the same nitrogen application rate (independent-sample *t*-test, $p < 0.05$). N1, 0 kg N·ha$^{-1}$; N2, 195 kg N·ha$^{-1}$; N3, 270 kg N·ha$^{-1}$; N4, 345 kg N·ha$^{-1}$. CV, the coefficient of variation. SG, the superior grain; IG, the inferior grain.

### 3.4. Cooking and Eating Quality

In contrast to N1, the hardness of nitrogen fertilizer treatments (N2, N3, N4) was higher, while the amylose content, viscosity, balance, taste value and gel consistency were lower (Table 4). The amylose content of N4 was 9.20–16.13% lower than N1, across 2019 and 2020 (Table 4). In comparison with N1, the taste value of SG and IG under N4 on average decreased by 14.1–24.9% and 20.1–25.1%, respectively, in 2019 and 2020. The amylose content of SG was 3.1–9.7% higher than that of IG (Table 4). The hardness of SG of the two varieties was poor than that of IG, and other eating quality related indicators were greater than those of IG in 2019 and 2020. The taste values of SG were 7.1–13.4% and 8.9–20.2% higher than those of IS, respectively, across 2019 and 2020. Except that the CV of hardness in SG was greater than that in IG, the CV of taste quality related indexes in IG was greater than that in SG. (Table 4). Increasing nitrogen application significantly reduced gel consistency. Compared with N1, the gel consistency of SG and IG under N4 decreased by 17.4–26.08% and 16.9–27.30%, respectively. In 2019 and 2020, the gel consistency of SG was 7.1–13.4% and 8.9–20.2% higher than that of IS, respectively (Table 4).

**Table 4.** Effects of nitrogen fertilizer levels on cooking and eating quality of strong and weak rice grains in 2019 and 2020.

| Year | Variety | Category | Treatment | Amylose Content (%) | Hardness | Viscosity | Balance | Taste Value | Gel Consistency (mm) |
|---|---|---|---|---|---|---|---|---|---|
| 2019 | NJ9108 | SG | N1 | 11.2 ± 0.35 Aa | 5.1 ± 0.15 Ac | 9.2 ± 0.05 Aa | 9.1 ± 0.10 Aa | 87.8 ± 0.62 Aa | 90.0 ± 1.00 Aa |
| | | | N2 | 10.8 ± 0.56 Aab | 5.4 ± 0.05 Bb | 8.8 ± 0.2 Aab | 8.7 ± 0.15 Aab | 84.8 ± 1.47 Aa | 87.0 ± 3.61 Aa |
| | | | N3 | 10.2 ± 0.13 Abc | 5.7 ± 0 Aa | 8.6 ± 0 Abc | 8.5 ± 0.05 Ab | 81.9 ± 0.38 Ab | 81.7 ± 4.04 Ab |
| | | | N4 | 9.7 ± 0.41 Ac | 5.9 ± 0.11 Ba | 8.2 ± 0.25 Ac | 8.0 ± 0.23 Ac | 79.2 ± 0.95 Ac | 74.3 ± 1.15 Ac |
| | | | CV (%) | 6.47 | 5.58 | 4.96 | 5.33 | 4.24 | 8 |
| | | IG | N1 | 10.7 ± 0.30 Aa | 5.1 ± 0 Ad | 9.2 ± 0.05 Aa | 9.1 ± 0.05 Aa | 87.7 ± 0.32 Aa | 84.7 ± 1.53 Ba |
| | | | N2 | 10.0 ± 1.21 Aab | 5.6 ± 0.05 Ac | 8.7 ± 0.3 Ab | 8.5 ± 0.25 Ab | 83.3 ± 2.60 Ab | 80.3 ± 1.15 Bb |
| | | | N3 | 9.8 ± 0.29 Bb | 5.8 ± 0.1 Ab | 8.0 ± 0.30 Bc | 7.9 ± 0.30 Bc | 78.3 ± 1.97 Bc | 76.0 ± 1.00 Bc |
| | | | N4 | 9.4 ± 0.15 Ab | 6.1 ± 0.15 Aa | 7.4 ± 0.75 Bd | 7.3 ± 0.52 Ad | 74.8 ± 3.31 Bd | 70.3 ± 1.15 Bd |
| | | | CV (%) | 7.36 | 7.07 | 9.46 | 9.07 | 6.72 | 7.24 |
| | NJ7 | SG | N1 | 17.9 ± 0.82 Aa | 6.0 ± 0.28 Ac | 8.5 ± 0.49 Aa | 8.2 ± 0.63 Aa | 80.1 ± 3.99 Aa | 81.3 ± 2.52 Aa |
| | | | N2 | 16.3 ± 0.22 Ab | 6.5 ± 0.05 Ab | 7.4 ± 0.23 Ab | 6.8 ± 0.15 Ab | 71.9 ± 0.92 Ab | 72.3 ± 2.89 Ab |
| | | | N3 | 15.7 ± 0.53 Abc | 6.9 ± 0.1 Ba | 5.8 ± 0.11 Ac | 5.6 ± 0.05 Ac | 63.7 ± 0.26 Ac | 68.0 ± 1.73 Ac |
| | | | N4 | 15.0 ± 0.43 Ac | 7.0 ± 0.20 Ba | 5.5 ± 0.09 Ac | 5.3 ± 0.23 Ac | 62.1 ± 1.48 Ac | 60.0 ± 2.65 Ad |
| | | | CV (%) | 7.42 | 6.59 | 18.8 | 18.77 | 10.97 | 11.81 |
| | | IG | N1 | 16.3 ± 0.36 Ba | 6.3 ± 0.20 Ac | 7.7 ± 0.25 Ba | 7.3 ± 0.30 Aa | 74.5 ± 1.84 Ba | 72.0 ± 1.00 Ba |
| | | | N2 | 15.5 ± 0.40 Bb | 6.8 ± 0.15 Ab | 6.7 ± 0.35 Bb | 6.2 ± 0.35 Bb | 68.1 ± 2.30 Bb | 63.7 ± 0.58 Bb |
| | | | N3 | 14.5 ± 0.23 Bc | 7.3 ± 0.05 Aa | 5.1 ± 0.1 Bc | 4.7 ± 0.15 Bc | 58.3 ± 0.80 Bc | 59.0 ± 1.00 Bc |
| | | | N4 | 14.2 ± 0.06 Bc | 7.3 ± 0.15 Aa | 5.0 ± 0.36 Bc | 4.8 ± 0.36 Bc | 57.7 ± 0.95 Bc | 53.7 ± 1.53 Bd |
| | | | CV (%) | 6.05 | 6.71 | 19.49 | 19.78 | 11.5 | 11.42 |
| 2020 | NJ9108 | SG | N1 | 11.1 ± 0.40 Aa | 5.0 ± 0.20 Bc | 9.2 ± 0.35 Aa | 9.2 ± 0.3 Aa | 87.9 ± 1.75 Aa | 84.7 ± 6.43 Aa |
| | | | N2 | 10.6 ± 0.06 Ab | 5.4 ± 0.26 Bbc | 8.4 ± 0.28 Ab | 8.6 ± 0.09 Ab | 83.0 ± 1.01 Ab | 78.7 ± 9.02 Aab |
| | | | N3 | 9.9 ± 0.05 Ac | 5.7 ± 0.25 Bb | 7.7 ± 0.20 Ac | 7.6 ± 0.26 Ac | 77.9 ± 2.03 Ac | 74.7 ± 1.15 Aab |
| | | | N4 | 9.6 ± 0.15 Ac | 6.3 ± 0.1 Ba | 7.2 ± 0.35 Ad | 7.1 ± 0.25 Ac | 73.8 ± 0.66 Ad | 67.0 ± 4.36 Ab |
| | | | CV (%) | 6.02 | 9.09 | 9.98 | 10.74 | 7.05 | 11.05 |
| | | IG | N1 | 9.9 ± 0.10 Ba | 5.6 ± 0.11 Ab | 8.7 ± 0.10 Ba | 8.6 ± 0.11 Ba | 84.5 ± 0.25 Ba | 73.3 ± 5.03 Ba |
| | | | N2 | 9.5 ± 0.15 Bb | 5.7 ± 0.11 Ab | 8.4 ± 0.17 Aa | 8.3 ± 0.10 Ba | 81.9 ± 1.01 Ab | 65.3 ± 1.53 Bb |
| | | | N3 | 9.1 ± 0.20 Bc | 6.1 ± 0.32 Aa | 7.3 ± 0.37 Bb | 7.4 ± 0.32 Ab | 73.7 ± 1.20 Bc | 60.7 ± 0.58 Bb |
| | | | N4 | 9.0 ± 0.24 Bc | 6.5 ± 0.20 Aa | 6.5 ± 0.3 Bc | 6.6 ± 0.32 Ac | 64.9 ± 0.96 Bd | 54.7 ± 1.15 Bc |
| | | | CV (%) | 4.28 | 6.9 | 12.16 | 10.78 | 10.56 | 11.79 |
| | NJ7 | SG | N1 | 18.7 ± 0.37 Aa | 5.9 ± 0.11 Bc | 7.7 ± 0.65 Aa | 7.9 ± 0.1 Aa | 78.1 ± 3.20 Aa | 72.7 ± 2.31 Aa |
| | | | N2 | 17.3 ± 0.19 Ab | 6.7 ± 0.26 Ab | 6.5 ± 0.26 Ab | 6.9 ± 0.4 Ab | 74.0 ± 2.82 Ab | 67.7 ± 1.53 Ab |
| | | | N3 | 16.6 ± 0.43 Ac | 6.9 ± 0.17 Ab | 5.7 ± 0.4 Abc | 6.2 ± 0.28 Ac | 68.3 ± 1.07 Ab | 62.7 ± 2.31 Ac |
| | | | N4 | 16.3 ± 0.10 Ac | 7.3 ± 0.23 Aa | 5 ± 0.17 Ac | 5.4 ± 0.2 Ad | 62.0 ± 2.41 Ac | 57.7 ± 0.58 Ad |
| | | | CV (%) | 5.67 | 8.1 | 17.77 | 14.82 | 9.44 | 9.27 |
| | | IG | N1 | 17.9 ± 0.11 Ba | 6.6 ± 0.35 Ac | 6.1 ± 1.70 Aa | 6.8 ± 1.35 Aa | 73.6 ± 1.78 Aa | 69.7 ± 2.08 Aa |
| | | | N2 | 16.9 ± 0.24 Ab | 6.9 ± 0.15 Abc | 5.6 ± 0.15 Bab | 5.5 ± 0.11 Bab | 67.2 ± 0.81 Bb | 63.0 ± 1.73 Bb |
| | | | N3 | 16.3 ± 0.51 Ab | 7.1 ± 0.15 Aab | 4.9 ± 0.15 Bab | 5.4 ± 0.28 Bab | 59.9 ± 1.25 Bc | 56.3 ± 1.53 Bc |
| | | | N4 | 15.5 ± 0.29 Bc | 7.6 ± 0.11 Aa | 4.1 ± 0.11 Bb | 4.5 ± 0.40 Bb | 55.2 ± 2.76 Bd | 50.7 ± 2.08 Bd |
| | | | CV (%) | 5.76 | 5.6 | 20.77 | 18.51 | 11.67 | 12.7 |

Different lowercase letters indicate statistically significant differences among nitrogen application rates according to a Duncan's multiple range test ($p < 0.05$). Different capital letters indicate statistical significance for the comparison between superior and inferior grains under the same nitrogen application rate (independent-sample $t$-test, $p < 0.05$). N1, 0 kg N·ha$^{-1}$; N2, 195 kg N·ha$^{-1}$; N3, 270 kg N·ha$^{-1}$; N4, 345 kg N·ha$^{-1}$. CV, the coefficient of variation. SG, the superior grain; IG, the inferior grain.

### 3.5. Protein Component Content

Protein component content of SG and IG increased gradually with the increase in nitrogen application rate (Table 5). Compared with N1, the contents of albumin, globulin, gliadin and glutenin increased by 10.78–59.02%, 14.55–47.23%, 9.82–32.62% and 21.98–48.42% under N4, respectively, in 2019 and 2020 (Table 5). In 2019 and 2020, protein component and total protein content of SG were lower than those of IG (except the albumin content of SG of Ningjing7 in 2019 was higher than that of IG). In addition, the CV of protein component content in SG was less than that in IG, which indicated that the effect of nitrogen application rate on protein component and total protein content in IG was greater than that in SG.

**Table 5.** Effects of nitrogen fertilizer levels on nutritional quality of superior and inferior grains in rice in 2019 and 2020.

| Year | Variety | Category | Treatment | Albumin (mg/g) | Globulin (mg/g) | Gliadin (mg/g) | Glutelin (mg/g) | Total Protein (mg/g) |
|---|---|---|---|---|---|---|---|---|
| 2019 | NJ9108 | SG | N1 | 2.81 ± 0.07 Ac | 3.30 ± 0.03 Ad | 4.63 ± 0.03 Ad | 51.6 ± 1.25 Ad | 62.4 ± 1.12 Ad |
| | | | N2 | 2.87 ± 0.01 Bbc | 3.52 ± 0.04 Bc | 5.10 ± 0.01 Ac | 54.0 ± 0.41 Bc | 65.5 ± 0.46 Bc |
| | | | N3 | 2.95 ± 0.06 Ab | 3.66 ± 0.03 Bb | 5.39 ± 0.57 Ab | 58.7 ± 0.85 Bb | 70.7 ± 1.21 Bb |
| | | | N4 | 3.27 ± 0.03 Ba | 3.78 ± 0.01 Ba | 5.76 ± 0.07 Aa | 62.9 ± 1.52 Ba | 75.8 ± 1.43 Ba |
| | | | CV (%) | 6.48 | 5.26 | 9.47 | 8.18 | 7.89 |
| | | IG | N1 | 2.34 ± 0.09 Bc | 3.07 ± 0.06 Bd | 4.23 ± 0.00 Bc | 53.3 ± 0.42 Ad | 63.0 ± 0.49 Ad |
| | | | N2 | 3.10 ± 0.03 Ab | 3.69 ± 0.04 Ac | 5.09 ± 0.06 Ab | 57.0 ± 0.30 Ac | 68.9 ± 0.25 Ac |
| | | | N3 | 3.12 ± 0.09 Ab | 4.08 ± 0.08 Ab | 5.54 ± 0.01 Aa | 60.4 ± 1.11 Ab | 73.2 ± 1.24 Ab |
| | | | N4 | 3.34 ± 0.01 Aa | 4.52 ± 0.02 Aa | 5.61 ± 0.03 Aa | 65.5 ± 0.62 Aa | 79.0 ± 0.65 Aa |
| | | | CV (%) | 13.34 | 14.62 | 11.24 | 7.98 | 8.65 |
| | NJ7 | SG | N1 | 2.35 ± 0.02 Ac | 2.71 ± 0.02 Ac | 4.49 ± 0.10 Bc | 51.0 ± 0.79 Bd | 60.6 ± 0.85 Bd |
| | | | N2 | 2.65 ± 0.09 Bb | 2.85 ± 0.04 Bb | 4.54 ± 0.01 Bc | 54.9 ± 1.21 Bc | 65.0 ± 1.12 Bc |
| | | | N3 | 2.66 ± 0.07 Bb | 2.92 ± 0.12 Bb | 4.82 ± 0.03 Bb | 62.2 ± 0.64 Bb | 72.7 ± 0.73 Bb |
| | | | N4 | 2.79 ± 0.07 Aa | 3.36 ± 0.06 Ba | 5.41 ± 0.09 Ba | 67.7 ± 1.16 Ba | 79.3 ± 1.19 Ba |
| | | | CV (%) | 6.8 | 8.78 | 8.03 | 11.53 | 10.87 |
| | | IG | N1 | 2.33 ± 0.04 Ab | 2.75 ± 0.11 Ad | 4.84 ± 0.00 Ac | 52.2 ± 1.26 Ad | 62.2 ± 1.14 Ad |
| | | | N2 | 2.63 ± 0.02 Ba | 2.99 ± 0.10 Ac | 4.95 ± 0.01 Ac | 58.2 ± 1.10 Ac | 68.8 ± 1.04 Ac |
| | | | N3 | 2.67 ± 0.02 Ba | 3.22 ± 0.05 Ab | 5.41 ± 0.09 Ab | 65.6 ± 2.41 Ab | 77.0 ± 2.50 Ab |
| | | | N4 | 2.72 ± 0.04 Aa | 3.50 ± 0.06 Aa | 5.73 ± 0.02 Aa | 70.9 ± 1.70 Aa | 82.9 ± 1.76 Aa |
| | | | CV (%) | 6.23 | 9.67 | 7.2 | 12.25 | 11.49 |
| 2020 | NJ9108 | SG | N1 | 2.82 ± 0.04 Ac | 3.12 ± 0.08 Ac | 4.41 ± 0.03 Ab | 42.4 ± 0.89 Bd | 52.8 ± 0.85 Bd |
| | | | N2 | 2.97 ± 0.03 Bb | 3.49 ± 0.09 Ab | 4.46 ± 0.28 Ab | 51.4 ± 0.77 Bc | 62.3 ± 0.88 Bc |
| | | | N3 | 3.08 ± 0.03 Bb | 3.62 ± 0.06 Aab | 4.94 ± 0.03 Aa | 56.4 ± 0.68 Bb | 68.1 ± 0.69 Bb |
| | | | N4 | 3.21 ± 0.08 Aa | 3.71 ± 0.07 Aa | 4.95 ± 0.03 Ba | 62.9 ± 1.19 Ba | 74.9 ± 1.16 Ba |
| | | | CV (%) | 5.18 | 7.02 | 6.28 | 14.77 | 13.15 |
| | | IG | N1 | 2.05 ± 0.02 Bc | 3.06 ± 0.04 Ac | 4.08 ± 0.06 Bd | 47.5 ± 1.26 Ad | 56.7 ± 1.39 Ad |
| | | | N2 | 3.01 ± 0.07 Ab | 3.45 ± 0.04 Ab | 4.64 ± 0.02 Ac | 57.4 ± 0.24 Ac | 68.6 ± 0.31 Ac |
| | | | N3 | 3.19 ± 0.02 Aa | 3.53 ± 0.02 Ab | 4.79 ± 0.03 Ab | 60.8 ± 0.50 Ab | 72.4 ± 0.57 Ab |
| | | | N4 | 3.26 ± 0.04 Aa | 3.73 ± 0.12 Aa | 5.27 ± 0.04 Aa | 65.1 ± 0.70 Aa | 77.4 ± 0.79 Aa |
| | | | CV (%) | 17.64 | 7.59 | 9.5 | 11.84 | 11.66 |
| | NJ7 | SG | N1 | 2.32 ± 0.01 Bb | 2.86 ± 0.08 Ac | 4.48 ± 0.01 Bc | 52.6 ± 0.47 Bd | 62.4 ± 0.51 Bd |
| | | | N2 | 2.34 ± 0.04 Bb | 3.15 ± 0.04 Ab | 4.65 ± 0.03 Ab | 56.2 ± 0.89 Ac | 66.4 ± 0.92 Ac |
| | | | N3 | 2.37 ± 0.04 Bb | 3.44 ± 0.04 Aa | 4.91 ± 0.00 Ba | 61.6 ± 0.82 Bb | 72.4 ± 0.80 Bb |
| | | | N4 | 2.57 ± 0.03 Ba | 3.49 ± 0.04 Aa | 4.92 ± 0.01 Ba | 66.8 ± 1.85 Ba | 77.9 ± 1.85 Ba |
| | | | CV (%) | 4.47 | 8.3 | 4.11 | 9.6 | 8.92 |
| | | IG | N1 | 2.37 ± 0.00 Ac | 2.99 ± 0.04 Ac | 4.65 ± 0.02 Ac | 54.2 ± 1.11 Ac | 64.3 ± 1.18 Ac |
| | | | N2 | 2.58 ± 0.12 Ab | 3.07 ± 0.07 Ac | 4.71 ± 0.05 Ac | 56.0 ± 1.42 Ac | 66.5 ± 1.46 Ac |
| | | | N3 | 2.81 ± 0.04 Aa | 3.39 ± 0.07 Ab | 5.05 ± 0.05 Ab | 66.3 ± 1.37 Ab | 77.7 ± 1.39 Ab |
| | | | N4 | 2.91 ± 0.02 Aa | 3.51 ± 0.05 Aa | 5.30 ± 0.02 Aa | 71.3 ± 1.99 Aa | 83.1 ± 1.96 Aa |
| | | | CV (%) | 8.51 | 7.1 | 5.65 | 12.15 | 11.31 |

Different lowercase letters indicate statistically significant differences among nitrogen application rates according to a Duncan's multiple range test ($p < 0.05$). Different capital letters indicate statistical significance for the comparison between superior and inferior grains under the same nitrogen application rate (independent-sample $t$-test, $p < 0.05$). N1, 0 kg N·ha$^{-1}$; N2, 195 kg N·ha$^{-1}$; N3, 270 kg N·ha$^{-1}$; N4, 345 kg N·ha$^{-1}$. CV, the coefficient of variation. SG, the superior grain; IG, the inferior.

### 3.6. Grain Filling Characteristics of Superior Grain (SG) and Inferior Grain (IG)

With the increase in nitrogen application rate, the grain weight of the two varieties decreased gradually, and the grain weight in superior grains was higher than in the inferior grains (Figure 2A,C). Compared with N1, the $G_{max}$ and $G_{mean}$ of SG under N4 decreased by 23.78–25.7% and 12.9–25.8%, while IG decreased by 17.7–24.0% and 12.7–20.3%, respectively. Increasing nitrogen fertilizer increased $T_{G.max}$ and $D$ of SG and IG (Figure 2B and Table 6).

Under excessive nitrogen application (N4), the $T_{G.max}$ and $D$ of SG of the two varieties were 32.9–39.8% and 27.5–29.4% higher than those under N1, while IG was 36.7–46.2% and 42.7–42.9% higher than those under N1, respectively (Figure 2D and Table 6). With the increase in nitrogen fertilizer level, the initial grain-filling rate ($R_0$) of SG and IG showed a decreasing trend. Compared with N1, $R_0$ of SG and IG under nitrogen fertilizer treatments decreased on average by 49.1–55.8% and 12.7–29.4%, respectively (Table 6).

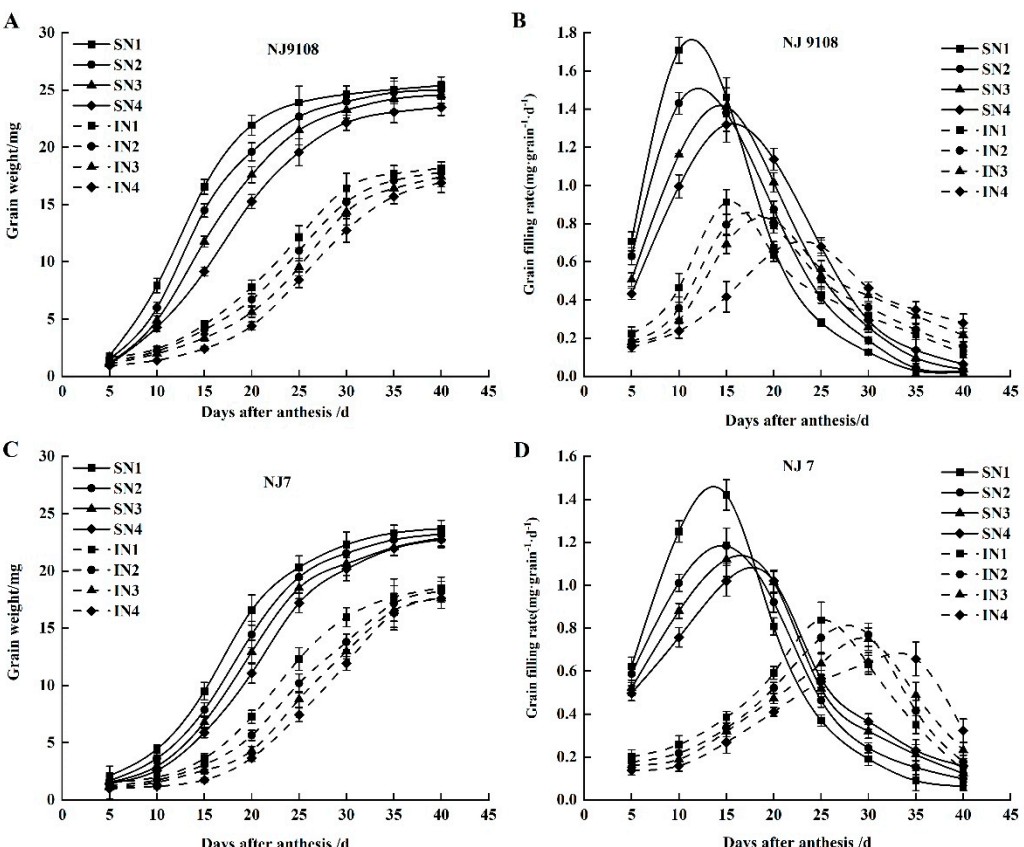

**Figure 2.** Grain weight (**A**,**C**) and grain filling rate (**B**,**D**) of superior (solid lines) and inferior (dotted lines) spikelets of rice. SN1 and IN1 represent the superior grain and inferior grain under N1 (0 kg N·ha$^{-1}$) condition, respectively. SN2 and IN2 represent the superior grain and inferior grain under N2 (195 kg N·ha$^{-1}$) condition, respectively. SN3 and IN3 represent the superior grain and inferior grain under N3 (270 kg N·ha$^{-1}$) condition, respectively. SN4 and IN4 represent the superior grain and inferior grain under N4 (345 kg N·ha$^{-1}$) condition, respectively.

### 3.7. Relationship between Rice Grain Quality and Grain Filling Characteristics

The correlation analysis indicated that the processing quality of SG had an extremely significant negative correlation with $R_0$, while the processing quality of IG was significantly negatively correlated with $G_{max}$ and significantly positively correlated with $D$. The appearance quality of SG and IG was significantly negatively correlated with $T_{G.max}$ and $D$, respectively, and was extremely significantly positively correlated with $G_{max}$ and $G_{mean}$, respectively (Table 7). The eating quality of SG and IG was extremely significantly negatively correlated with $T_{G.max}$ and $D$, and significantly positively correlated with $G_{max}$, $G_{mean}$ and $R_0$. The protein component content in SG and IG was significantly positively correlated with $T_{G.max}$ and $D$, and extremely significantly negatively correlated with $G_{max}$ (Table 7).

**Table 6.** Parameters of grain-filling characteristics of superior and inferior grains.

| Variety | Category | Treatment | $T_{G.max}$/d | $G_{max}$/ (mg·Grain$^{-1}$·d$^{-1}$) | $G_{mean}$/ (mg·Grain$^{-1}$·d$^{-1}$) | D/d | $R_0$ |
|---|---|---|---|---|---|---|---|
| NJ9108 | SG | N1 | 11.5 ± 0.58 Bd | 1.8 ± 0.09 Aa | 1.2 ± 0.06 Aa | 20.8 ± 0.62 Bd | 0.7 ± 0.03 Aa |
| | | N2 | 12.3 ± 0.42 Bc | 1.5 ± 0.05 Ab | 1.1 ± 0.04 Ab | 23.3 ± 0.93 Bc | 0.4 ± 0.01 Ab |
| | | N3 | 14.0 ± 0.56 Bb | 1.4 ± 0.06 Abc | 1.0 ± 0.04 Ac | 25.4 ± 0.51 Bb | 0.4 ± 0.01 Ab |
| | | N4 | 16.1 ± 0.32 Ba | 1.3 ± 0.03 Ac | 0.9 ± 0.02 Ad | 26.5 ± 0.79 Ba | 0.3 ± 0.02 Ac |
| | | CV (%) | 15 | 13 | 13 | 10.5 | 41.1 |
| | IG | N1 | 15.2 ± 0.76 Ad | 0.9 ± 0.05 Ba | 0.6 ± 0.03 Ba | 28.9 ± 0.64 Ad | 0.2 ± 0.01 Bb |
| | | N2 | 17.9 ± 0.90 Ac | 0.9 ± 0.04 Bb | 0.6 ± 0.03 Bab | 33.1 ± 0.73 Ac | 0.3 ± 0.02 Ba |
| | | N3 | 19.8 ± 0.79 Ab | 0.8 ± 0.03 Bb | 0.6 ± 0.02 Bb | 37.2 ± 0.74 Ab | 0.1 ± 0.01 Bc |
| | | N4 | 22.2 ± 0.44 Aa | 0.7 ± 0.01 Bc | 0.5 ± 0.01 Bc | 41.3 ± 0.91 Aa | 0.1 ± 0.01 Bc |
| | | CV (%) | 15.8 | 11.3 | 9.3 | 15.2 | 47.9 |
| NJ7 | SG | N1 | 13.7 ± 0.69 Bc | 1.4 ± 0.07 Aa | 0.9 ± 0.04 Aa | 28.7 ± 1.43 Ad | 0.6 ± 0.03 Aa |
| | | N2 | 14.2 ± 0.53 Bc | 1.2 ± 0.04 Ab | 0.8 ± 0.03 Aab | 29.8 ± 0.95 Bc | 0.3 ± 0.02 Ab |
| | | N3 | 16.2 ± 0.65 Bb | 1.2 ± 0.05 Abc | 0.8 ± 0.03 Abc | 31.3 ± 0.97 Bb | 0.3 ± 0.01 Ab |
| | | N4 | 18.2 ± 0.36 Ba | 1.1 ± 0.02 Ac | 0.7 ± 0.01 Ac | 33.2 ± 1.06 Ba | 0.2 ± 0.01 Ac |
| | | CV (%) | 13.2 | 12.2 | 6.9 | 6.4 | 50.9 |
| | IG | N1 | 24.7 ± 0.49 Ad | 0.8 ± 0.02 Ba | 0.6 ± 0.01 Ba | 31.5 ± 0.53 Ad | 0.2 ± 0.01 Ba |
| | | N2 | 27.6 ± 0.94 Ac | 0.8 ± 0.03 Ba | 0.6 ± 0.02 Ba | 36.5 ± 1.21 Ac | 0.2 ± 0.01 Bb |
| | | N3 | 29.9 ± 0.72 Ab | 0.8 ± 0.02 Bb | 0.5 ± 0.01 Bb | 40.8 ± 0.98 Ab | 0.1 ± 0.01 Bc |
| | | N4 | 33.8 ± 0.68 Aa | 0.7 ± 0.01 Bc | 0.5 ± 0.01 Bb | 43.9 ± 0.88 Aa | 0.1 ± 0.01 Bd |
| | | CV (%) | 13.2 | 8.5 | 6.4 | 14.1 | 24.9 |

$T_{G.max}$, the date of reaching the maximum grain filling rate; $G_{max}$, the maximum grain filling rate; $G_{mean}$, the average grain filling rate; $D$, the active grain filling period; $R_0$, initial grain-filling rate. SG, the superior grains, IG, the inferior grains. Different lowercase letters indicate statistically significant differences among nitrogen application rates according to a Duncan's multiple range test ($p < 0.05$). Different capital letters indicate statistical significance for the comparison between superior and inferior grains under the same nitrogen application rate (independent-sample $t$-test, $p < 0.05$). N1, 0 kg N·ha$^{-1}$; N2, 195 kg N·ha$^{-1}$; N3, 270 kg N·ha$^{-1}$; N4, 345 kg N·ha$^{-1}$. CV, the coefficient of variation.

**Table 7.** Correlation of rice grain-filling parameters with grain quality.

| | Superior Grain | | | | | Inferior Grain | | | | |
|---|---|---|---|---|---|---|---|---|---|---|
| | $T_{G.max}$ | $G_{max}$ | $G_{mean}$ | $D$ | $R_0$ | $T_{G.max}$ | $G_{max}$ | $G_{mean}$ | $D$ | $R_0$ |
| Brown rice rate | 0.760 * | −0.674 | −0.560 | 0.588 | −0.909 ** | 0.239 | −0.683 | −0.453 | 0.787 * | −0.393 |
| Milled rice rate | 0.695 | −0.542 | −0.430 | 0.448 | −0.868 ** | 0.135 | −0.747 * | −0.494 | 0.748 * | −0.458 |
| Head rice rate | 0.820 * | −0.707 | −0.604 | 0.633 | −0.930 ** | 0.215 | −0.787 * | −0.553 | 0.792 * | −0.471 |
| Chalkiness rate | −0.825 * | 0.920 ** | 0.926 ** | −0.891 ** | 0.690 | −0.952 ** | 0.661 | 0.844 ** | −0.682 | 0.701 |
| Chalkiness area | −0.817 * | 0.874 ** | 0.845 ** | −0.895 ** | 0.641 | −0.475 | 0.733 * | 0.654 | −0.830 * | 0.540 |
| Chalkiness degree | −0.825 * | 0.918 ** | 0.924 ** | −0.890 ** | 0.689 | −0.846 ** | 0.789 * | 0.871 ** | −0.834 * | 0.707 * |
| Grain length | 0.390 | −0.617 | −0.718 * | 0.678 | −0.037 | 0.687 | −0.076 | −0.386 | 0.044 | −0.249 |
| Grain width | −0.848 ** | 0.917 ** | 0.934 ** | −0.964 ** | 0.550 | −0.970 ** | 0.862 ** | 0.940 ** | −0.880 ** | 0.747 * |
| Length-width ratio | 0.591 | −0.764 * | −0.835 ** | 0.824 * | −0.242 | 0.970 ** | −0.571 | −0.783 * | 0.569 | −0.588 |
| Hardness | 0.893 ** | −0.963 ** | −0.939 ** | 0.993 ** | −0.678 | 0.995 ** | −0.748 * | −0.904 ** | 0.762 * | −0.687 |
| Viscosity | −0.900 ** | 0.932 ** | 0.872 ** | −0.982 ** | 0.704 | −0.986 ** | 0.790 * | 0.923 ** | −0.803 * | 0.737 * |
| Balance | −0.909 ** | 0.944 ** | 0.888 ** | −0.989 ** | 0.720 * | −0.986 ** | 0.780 * | 0.917 ** | −0.805 * | 0.712 * |
| Taste value | −0.923 ** | 0.947 ** | 0.902 ** | −0.988 ** | 0.714 * | −0.964 ** | 0.855 ** | 0.954 ** | −0.871 ** | 0.765 * |
| Gel consistency | −0.961 ** | 0.957 ** | 0.921 ** | −0.976 ** | 0.765 * | −0.916 ** | 0.922 ** | 0.957 ** | −0.951 ** | 0.727 * |
| Albumin | −0.035 | 0.297 | 0.422 | −0.376 | −0.300 | 0.011 | −0.570 | −0.359 | 0.585 | −0.147 |
| Globulin | 0.224 | 0.029 | 0.192 | −0.092 | −0.549 | −0.125 | −0.546 | −0.265 | 0.546 | −0.201 |
| Prolamin | 0.656 | −0.400 | −0.300 | 0.342 | −0.764 * | 0.636 | −0.943 ** | −0.853 ** | 0.936 ** | −0.609 |
| Glutelin | 0.967 ** | −0.844 ** | −0.766 * | 0.829 * | −0.884 ** | 0.708 * | −0.941 ** | −0.858 ** | 0.983 ** | −0.636 |
| Total protein | 0.940 ** | −0.791 * | −0.701 | 0.768 * | −0.895 ** | 0.655 | −0.941 ** | −0.837 ** | 0.978 ** | −0.613 |
| Amylose content | 0.307 | −0.536 | −0.666 | 0.592 | 0.068 | 0.743 * | −0.161 | −0.461 | 0.125 | −0.326 |

$T_{G.max}$, time reaching the maximum grain-filling rate; $G_{max}$, the maximum grain filling rate; $G_{mean}$, the average grain filling rate; $D$, the active grain filling period; $R_0$, initial grain-filling rate. * and **, significant at $p < 0.05$ and $p < 0.01$, respectively.

## 4. Discussion

### 4.1. Effect of Nitrogen Level on Rice Yield and Grain Filling

Nitrogen was an indispensable nutrient element in the growth and development of rice, which was closely related to the formation of rice yield [32]. Excessive nitrogen application reduced nitrogen use efficiency and rice yield [33,34]. Therefore, appropriate nitrogen application was of great significance in agricultural production and was imperative. In this two-year field experiment, we found that the yield of N3 was higher, which was mainly attributed to the higher panicle per m$^2$ and grain number per panicle. The seed setting rate and 1000-grain weight of N4 were significantly lower than that of N3 (Table 1), which

suggested that grain filling might be affected by excessive nitrogen treatment. Zhang et al. [7] found that nitrogen application decreased the grain-filling rate of the superior and inferior grains, postponing the maximum time of grain filling and prolonging the effective grain-filling duration. Our results were consistent with theirs (Figure 2 and Table 6), probably because the high nitrogen input increased nitrogen concentrations in plant tissue, which might bring about a high rate of nitrogen metabolism and result in the enhancement of carbohydrate consumption and reduction in carbohydrate translocation to grain filling [35]. Similarity, Zhu et al. [34] found that with the increasing nitrogen application level, grain yield of both varieties first increased and then decreased. Indeed, it was reported that higher nitrogen application rate significantly reduced the filling rate of SG, not IG, compared with a lower nitrogen application rate [36]. However, Chen et al. [37] found that the grain weight and grain-filling rate of inferior spikelets were significantly promoted by normal nitrogen application when compared with those of the high-nitrogen treatment. We also found the same results. In this study, the maximum grain filling rate and average grain filling rate decreased with the increase in nitrogen fertilizer level (Figure 2 and Table 6). There are similar findings in wheat; it was reported that high nitrogen application reduced the grain-filling rate and grain weight of IG in the middle spikelets compared with those under normal nitrogen treatment [38]. Chen et al. [37] found that excessive nitrogen application reduced the accumulation of cytokinin and auxin, accordingly inhibiting the filling of inferior grains. On the other hand, high nitrogen inhibited the sucrose transport and starch accumulation characteristics in rice, thereby reducing the grain filling rate [37,38].

It has been reported that cytokinin play important roles during grain filling, and are also closely linked to nitrogen signaling [39]. The relationship between cytokinins and nitrogen application has also been intensively studied in roots and leaves [40,41]. Low nitrogen was found to reduce cytokinin content by inhibiting the biosynthesis of cytokinin, and the application of moderate nitrogen increased the expression of cytokinin biosynthesis genes and thus promoted cytokinin content in roots and leaves [41,42]. However, many studies reported that CTK content was regulated by catabolism rather than biosynthesis in response to excessive nitrogen treatment [37,43]. The cytokinins were lower in inferior spikelets by high nitrogen application (240 kg N ha$^{-1}$), resulting in significantly reduced grain filling when compared with normal treatment (120 kg N ha$^{-1}$) [37]. The response of catabolism and anabolism of cytokinins in superior and inferior grains to nitrogen fertilizer needs to be further studied. Excessive nitrogen application affected the production, transportation, distribution and utilization of photosynthetic assimilates, which further influenced the grain filling process [44]. Therefore, appropriately reducing the nitrogen application rate can shorten the grain filling period of SG and IG, and finally increase the grain weight and yield. The physiological mechanism of poor grain filling of IS of *japonica* rice under excessive nitrogen application conditions will be the focus of further research.

*4.2. Differences in Grain Quality between SS and Is under Different Nitrogen Application Rate*

Many studies have shown that increasing nitrogen fertilizer can improve rice processing quality [7,34]. We also found the same rule (Table 2). It was worth mentioning that the indexes of processing quality of SG were better than that of IG (Table 2). This may be because the grain filling rate of SG was faster, while the grain filling rate of IG was slow. The SG can obtain sufficient nitrogen, which was beneficial to the synthesis and accumulation of grain protein and its components, which increased grain hardness. Rice grain appearance quality mainly includes chalky rate, chalkiness area, chalkiness and grain type. The increasing nitrogen level also decreased the appearance quality [34]. Our results were consistent with that, which suggested that excessive nitrogen application reduced the grain filling rate and grain appearance quality (Table 3). The possible reason was that the increasing nitrogen application increased the spikelets per m$^2$ and prolonged the grain filling period, resulting in low grain filling rate and poor grain filling, which eventually leaded to the loose arrangement of starch granules, resulting in increased grain chalkiness.

However, we found that the grain filling rate was significantly positively correlated with the appearance quality (Table 7), and Zhang et al. [7] also found similar results, in which the internal relationship and physiological mechanism need to be further proved. It has been reported that when the total nitrogen application rate was 300 kg ha$^{-1}$, the chalkiness area and chalkiness of SG and IG increased significantly compared with the CK, and the appearance quality became worse [45]. In this study, we found that the chalky rate, chalkiness area, chalkiness of the SS and IS were decrease in the range of 0–195 kg ha$^{-1}$ with the increase in nitrogen fertilizer application. However, the chalkiness increased slightly when the nitrogen application rate was more than 195 kg ha$^{-1}$. This may be because within a certain nitrogen fertilizer level, increasing nitrogen fertilizer could slow down the grain filling rate of SG and IG, prolong the grain filling period, make starch granules and protein bodies arranged closely and the gap smaller, thereby reducing chalkiness. The chalkiness degree was reported to be significantly negatively correlated with grain eating quality, which may be because high chalkiness means that the density of starch particles is low, and starch particles are easier to break during cooking [46].

The cooking and eating quality is the core index for evaluating grain quality, which mainly includes amylose content, gel consistency and taste value. In this study, we found that excessive nitrogen application reduced amylose content, gel consistency, and taste value of SS and IS, and increased the cooked rice hardness (Table 4), which seemed to have a significantly positive correlation agreeing with the study reported by Zhu et al. [34]. However, many studies had reported a negative correlation between amylose content and taste value [47], the possible reason was that the cooking quality was carried out by satake rice taste analyzer, which is mainly affected by amylose and protein content. Although nitrogen application decreased the amylose content and increased the starch quality, excessive nitrogen application significantly increased the protein content and finally decreased the eating and cooking quality. The CV of taste quality related indexes shows that the effect of nitrogen application rate on IG was greater than that on SG. Moreover, amylose is moderated by both genes and the environment [48]. It is reported that high ADP-glucose pyrophosphorylase (AGPase) and starch branching enzyme (SBE) activities in grains will promote the conversion of amylose to amylopectin [46]. Increasing nitrogen application could improve the enzyme activities of ADPase and SBE in SG and IG, thereby increasing the amylopectin content and reducing the amylose content. Compared with SG, the amylose content of IG was lower; we speculated that the possible reason was that high protein content in SG inhibited the expansion of starch granules.

Increasing nitrogen application can increase grain protein content [7,49,50]. In our study, we found that the protein content of SG and IG increased with the increase in the nitrogen application rate, and the protein content of IG was much higher than that of SG (Table 5). This may be because the increase in nitrogen application promotes the transfer of nitrogen from roots to grains, and increased the accumulation of protein components in grains, especially the accumulation of glutenin. We found that glutenin was significantly negatively correlated with grain filling rate (Table 7), and the physiological mechanism behind this relationship needs to be further revealed. In addition, IG still had certain filling ability at the late grain filling stage, and sufficient nitrogen supply might increase the volume and number of proteosomes of the IG at the end of grain filling stage. The nitrogen application rate was positively correlated with protein content when the nitrogen fertilizer level was in the range of 60–180 kg ha$^{-1}$ [49,50]. Higher yield and better rice quality are the goals pursued by agricultural scientists and technicians, but there are often contradictions between them [51,52]. The research on nitrogen fertilizer application technology that takes into account both yield and quality will be the focus of future work. Furthermore, at the late grain filling stage, the volume and number of protein bodies in SG decreased rapidly, while the volume and number of protein bodies in IG remained at a low level and increased [53], which eventually led to the high protein content in IG. In addition, NJ9108 and NJ7 are super rice varieties, and had the same response to N fertilizer level on grain yield and grain quality parameters. Under low nitrogen application rate conditions, the yield of NJ7 was

slightly higher than that of NJ9108, indicating that NJ7 was sensitive to nitrogen fertilizer. It is speculated that the nitrogen use efficiency of the two varieties is quite different, which will be the focus of the next study. It is worth mentioning that the rice quality of NJ9108, especially the eating quality, is better than that of NJ7, and NJ9108 may be more popular with farmers and consumers.

## 5. Conclusions

Increased nitrogen application can promote the grain yield of rice in a certain range. Excessive nitrogen application reduced the seed setting rate and 1000-grain weight, resulting in a lower yield. Excessive nitrogen application reduced the maximum and average grain filling rate of SG and IG of *japonica* rice, and prolonged the active period of grain filling. Excessive nitrogen application weakened the grain quality of SG and IG. Through two years of field experiments, we found that with the increase in nitrogen fertilizer level, the cooking and eating quality of SG and IG of *japonica* rice became worse, and the processing quality, appearance quality and nutritional quality were improved to a certain extent. The grain quality of SG was better than that of IG, and the indexes of grain quality in IS were more sensitive to nitrogen fertilizer. Our results suggest that optimal nitrogen application may improve grain yield and grain quality by regulating the grain filling characteristics of superior and inferior grains.

**Author Contributions:** Conceptualization, Z.H.; methodology, C.Z.; software, C.Z. and G.L.; validation, G.L., Y.C. and Y.J.; formal analysis, C.Z. and Z.H.; investigation, G.L., Y.C. and Y.S.; re-sources, L.Z., P.L. and W.W.; data curation, C.Z. and G.L.; writing—original draft preparation, C.Z.; writing—review and editing, Z.H., K.X. and Q.D.; visualization, P.L. and W.W.; supervision, Z.H. and Q.D.; project administration, K.X. and Q.D.; funding acquisition, Z.H. All authors have read and agreed to the published version of the manuscript.

**Funding:** This study was finically supported by the National Natural Science Foundation of China (32001469), the Key Research Program of Jiangsu Province (BE2020319, BE2019377, BE2021361), the National Key Research and Development Program of China (2018YFD0300802), the National Rice Industrial Technology System (CARS-01-28), the Postgraduate Research & Practice Innovation Program of Yangzhou University (KYCX21_3242), and the Priority Academic Program Development of Jiangsu Higher Education Institutions (PAPD).

**Institutional Review Board Statement:** Not applicable.

**Data Availability Statement:** Data is contained within the article.

**Acknowledgments:** The authors thank Zhi Dou, Pinglei Gao and Rui Liu for their assistance with the experiments.

**Conflicts of Interest:** The authors declare that they have no known competing financial interest or personal relationships that could appear to influence the work reported in this paper.

## Abbreviations

SG, superior grains; IG, inferior grains; ENA, excessive nitrogen application; (ANOVAs), analyses of variance; (CV), coefficient of variation.

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
