# Peer review of "Excessive Nitrogen Application Leads to Lower Rice Yield and Grain Quality by Inhibiting the Grain Filling of Inferior Grains"

_agriculture, doi:10.3390/agriculture12070962_

Round 1
Reviewer 1 Report
Excessive nitrogen reduced the yield and grain quality of japonica rice by inhibiting the grain filling of inferior grain by Can Zhao , Guangming Liu , Yue Chen , Yan Jiang , Yi Shi , Lingtian Zhao , Pingqiang Liao , Weiling Wang , Zhongyang Huo * , Ke Xu , Qigen Dai tried a good attempt in rice physiology, but it needs more additional experiments. 1. nitrogen application has a strong relation with cytokinin metabolism. athours should proof cytokinin level,and its signaling components upregulate or down regulate in which nitrogen doses. it will give a clear picture of exprimentAuthor Response
Please see the attachment.

Reviewer 2 Report
Abstract
Line 12. The abbreviated letter should be used after following the full word. Such as ENA in Line 17 should be after excessive nitrogen application.
Line 19, what are SS and IS. Is it the rice variety name? Please be careful to use the abbreviation in the Abstract. It would be missed out understanding for readers.
Introduction
In line 37. The author mentioned that the excessive nitrogen used in rice field. To be more significant study on the excessive rate affecting rice production. Can the authors provide the optimum / recommend rate of nitrogen rate in China in the introduction part.
There are many symbols such as SG, IG, SS, and IS. Can the authors avoid using it in the introduction part? It would be easy to read. The full and abbreviation should be reported in the material and method part. Also it can be reported in graph or table in the result if that is needed.
Material and method
Line 97. The split pot design was used, what are the main plot and subplot. While the result part in table 1 showed three factors including year, variety and nitrogen treatment.
Results
According to the topic title. Subdivision should be started with rice yield and its components followed by grain quality, respectively.
Line 167. Could the authors provide the grain type (i.e. brown rice or milled rice)? Also, figure 1A (on the left side) and 1B (on the right side) should be specified in both text and figure to describe clearly.
Line 168. Please check the abbreviation, there are different in the text and figure 1 (SG and IG / S and I)
Subsection 3.2
The suggestion is to use technical term between seed setting rate or filled grain similarly in text and table 1.
Table 1. The data should be started with grain yield in the right column then followed by yield components. It would be enough significant to provide only one decimal number for panicles number and grains per panicle and filled grain. There are two decimal numbers for grain yiled, but single is reported. In addition, SE values should be provided in Table 1 and 2 same as Tables 3-6.
Please rewrite the text in 3.2. For example, grain yield
Line 186-187. Year and fertilizer significantly affected grain yield without interaction. The authors
The increase in N fertilizer level resulted in the increasing of grain yield, however the excessive N fertilizer rate (N4) was not significantly deceased grain yield compared to N3 for both varieties and both years (there is the same letter between N3 and N4). It is showing that the excessive N fertilizer did not strongly reduced grain yield, but it trended to be lower yield when N4 applied.
The interactions of variety and treatment and variety and year were highly significant. However, the report in Table 1 did not show the interaction effect. Also, the factor of N fertilizer was mainly described in the text. It would be more interesting to describe the interaction between N fertilizer and variety to see how much different of N fertilizer levels affecting yield component in two varieties. If the authors agree.
Table 2. If there is significant different, please provide the different letter.
Figure 2 is very clear. However, figure 2A and 2C had no description in the text.
Could the authors merge Table 3 and 4 to be the same table? And then describe the results of SG and IG of two varieties under different N application rates. This suggestion is similar as Table 5 and 6, Table 7 and 8 and Table 9 and 10.
Lines 230-231 and lines 289-290. Is there significant to describe the CV value of 2019 and 2020? If not, please delete.
Could the authors move subsection 3.1 grain morphology before 3.5 Appearance quality. Meanwhile, 3.1 and 3.5 can be combined to be the same subsection.
Line 256. Please check the year
Lines 260-262, 289-290 and 317-320. How can the author calculate %CV. Please check it again, it normally indicates a quality assurance of data.
Line 360. Collect the sentence, The seed setting rate and 1000-grain weight were reduced by excessive nitrogen application, resulting in a decrease in yield
Discussion
Rice variety is one of factors that used in the study. Could the author discuss the effect rice variety affecting by N fertilizer level? Even though two rice varieties had the same response to N fertilizer level on grain yield and grain quality parameters. It would be useful as a guide for farmers who grow other rice varieties.
Reviewer 3 Report
Introduction section has enough information about the subject of the study. Methodology is accurate and well-articulated. Results of the present research work are novel and contribute to the advancement of understanding and are pragmatic. Discussion is well written. In overall manuscript I will suggest that please check the needs for tables and figures and the adequacy of the references. Check the spellings. The present work is novel and well written.
Reviewer 4 Report
The authors have executed and reported a well designed study on effects of excessive N on grains originating at different positions on the spike.
Few of the observations which could be addressed for greater clarity
L64 Sg should be SG
L97-98 What was the basis of selection of these two cultivars? It is better to mention this here
L 108-110 The appropriate growth scale like Zadoks in wheat may be mentioned to indicate crop growth stages for clarity and repeatability
L159 Data analysis may be elaborated to specify the fixed and random effects of ANOVA
L173-174 Instead of just mentioning slender, this could be quantified with measurements (either weight or dimensions) in the parenthesis.
Results
Table shows that yield of N4 was not statistically inferior to N3 or other treatments, then how the authors have concluded that yield was reduced under excessive N? In title and also in abstract?
Round 2
Reviewer 1 Report
Excessive nitrogen reduced the yield and grain quality of japonica rice by inhibiting the grain filling of inferior grains by Can Zhao et al. is now accepted. auhors does not address to my point but iin future they will extend the work.